# One-Pot Processing of Regenerated Cellulose Nanoparticles/Waterborne Polyurethane Nanocomposite for Eco-friendly Polyurethane Matrix

**DOI:** 10.3390/polym11020356

**Published:** 2019-02-18

**Authors:** Soon Mo Choi, Min Woong Lee, Eun Joo Shin

**Affiliations:** 1School of Chemical Engineering, Yeungnam University, 280 Daehak-Ro, Gyeongsan, Gyeongbuk 38541, Korea; smchoi@ynu.ac.kr; 2Regional Research Institute for Fiber & Fashion Materials, Yeungnam University, 280 Daehak-Ro, Gyeongsan, Gyeongbuk 38541, Korea; 3Department of Organic Materials and Polymer Engineering, Dong-A University, 550-37 Nakdong-daero, Saha-gu, Busan 49315, Korea; tnlswkqjqtk@naver.com

**Keywords:** regenerated cellulose nanoparticles (RCNs), waterborne polyurethanes (WPU), nanocomposites, polyols, isocyanates, one-pot processing.

## Abstract

Regenerated cellulose nanoparticles (RCNs) reinforced waterborne polyurethanes (WPU) were developed to improve mechanical properties as well as biodegradability by using a facile, eco-friendly approach, and introducing much stronger chemical bonding than common physical bonding between RCNs and WPU. Firstly, RCNs which have an effect on improving the solubility and stability of a solution, thereby resulting in lower crystallinity, were fabricated by using a NaOH/urea solution. In addition, the stronger chemical bond between RCNs and WPU was here introduced by regarding at which stage in particular added RCNs worked best on strengthening their bond in the process of WPU synthesis. The chemical structure, mechanical, particle size and distribution, viscosity, and thermal properties of the resultant RCNs/WPU nanocomposites were investigated by Fourier transform infrared analysis (FTIR), Zeta-potential analysis, viscometer, thermogravimetric analysis (TGA), Instron, and dynamic mechanical analysis (DMA). The results of all characterizations indicated that the RCNs/WPU-DMF associated with the addition of RCNs in DMF-dispersed step resulted in more effectively crosslinked between WPU and nano-fillers of nanocellulose particles in the dispersion than Acetone and Water-dispersed steps, thereby attributing to novel interactions formed between RCNs and WPU.

## 1. Introduction

The reinforcement of nanoparticles within a continuous polymeric phase to improve the thermal and mechanical properties of resultant nanocomposite has been attracting attention in high functional applications such as electronics, tissue engineering, and civil [1,2,3,4]. In particular, the mechanical properties depend on the constituents of the nanocomposite as well as the phase morphology. Polyurethanes (PUs) consisting of both isocyanate (hard segments) and polyol (soft segments) are one of the polymers class with the widest applications by engineering polymer structure, thereby having various desirable properties. In spite of the strong advantage related to their versatile applicability, the use of organic solvents has been a controversial one to prepare PU-based composites. Consequently, waterborne polyurethanes (WPUs) have been replaced by conventional PUs associated with the use of organic solvents owing to growing concerns about environmental issues, health, toxicity, and the extended application caused by substitution of it. Because of global limitations on the release of volatile organic compounds (VOCs) into the atmosphere, eco-friendly products have gained popularity in diverse industries over recent years [5,6,7]. Viewed from this angle, water is the best medium in order to synthesize chemicals due to it being safe, nontoxic, and a very low in price. The synthesis process of WPU is conducted in water which functions not only as an emulsifier, but also solvent reacting with WPUs pre-polymer. Therefore, the WPU has been considered specific materials to a particular global condition. This is a group of multiblock copolymer elastomeric materials with alternating polyether or polyester soft segments and rigid urethane hard segments [8]. Due to the thermodynamic differences between the hard and soft segments, phase separation often occurs during cooling. The hard segment domain acts as a physical intersection point and as a filler with a higher modulus than the soft segment matrix. This network molecular structure exhibits excellent flexibility and elasticity. Nonetheless, the improvement of the low thermal stability and mechanical properties of WPU is still a matter of further discussion [9,10]. We here focused on the introduction of regenerated cellulose nanoparticles (RCNs) produced from microcrystalline cellulose (MCC) isolated from diverse natural resources in order to overcome the mentioned drawbacks. RCNs have received peculiar attention due to their unique features such as biodegradability, biocompatibility, renewability, light weight, high aspect ratio, and, especially, abundance [7,8]. However, many researchers have striven to concentrate on solving the biggest challenges about dispersing cellulose in aqueous solutions with no use of surfactant and/or modification. Accordingly, NaOH-based solutions and urea were introduced for improving both its solubility and the stability of solution. This mechanism can be explained by the strong interaction with NaOH to reduce the aggregation of cellulose molecules through the formation of new hydrogen bonds between cellulose and NaOH [7,11,12]. The improvement of its solubility in NaOH/urea aqueous solvents results in low degree of crystallinity. Also, the dispersion of RCNs within polymeric phase enables to enhance both low thermo-stability and biodegradability of polymeric phase, WPUs [13,14,15,16,17]. This strategy provides a potential for dispersing of RCNs into WPU through incorporating RCNs into the polyols which has a decisive influence on the the final WPU product characteristics [18]. We previously developed RCNs/WPU nanocomposites through this approach, and also investigated the effect of RCNs incorporation on their performance and properties. 

Furthermore, in this study, we strengthened the bond of RCNs with WPU matrix within the nanocomposite through the adding of RCNs in each process sequence of WPU synthesis, respectively. In addition, it was found what a particular stage added by RCNs worked best on strengthening the bond among all processes. Therefore, the final purpose is to develop high-stretchable WPU films with good biocompatibility by introducing not most of physical bond, but much stronger chemical bond between RCNs and WPUs.

## 2. Experimental

### 2.1. Materials

MCC, Polycaprolactone diol (*M*_n_ = 1250), isophorone diisocyanate (IPDI), dimethylol propionic acid (DMPA), and triethylamine (TEA) was purchased from Sigma-Aldrich (St. Louis, MO, USA) Corporation and used as received without further purification. Urea, NaOH, and other chemicals used were all AR grade. Celluase enzyme obtained from Aspergillusniger was purchased from Sigma. *N*,*N*-dimethylformamide (Alfa Aesar, Haverhill, MA, USA) and Acetone (JUNSEI, Tokyo, Japan) were used without further purification. 

### 2.2. Preparation of RCNs

The preparation of regenerated cellulose nanoparticles (RCNs) is described in previous paper [7], which were prepared from MCC through the method described by Adsul et al. [19]. Obtaining RCNs aqueous dispersion with solid contents was about 1 wt %. To compare properties of RCNs with commonly used cellulose nanocrystals (CNCs), the CNCs were obtained via acid hydrolysis of MCC. This process was already published several times by authors [20]. Briefly, the MCC and H_2_SO_4_(64 wt %) were mixed at 45 °C for 30 min. Hydrolysis was stopped by pouring cold deionized water, and the suspension was washed by centrifugation. The resulting suspension was dialyzed against deionized water to a pH about 8 to obtain a RNC aqueous dispersion having a solids content of about 0.5 wt %.

### 2.3. Preparation of WPU and RCNs/WPU Nanocomposites Solutions

The preparation of WPU and RCNs/WPU is also described in previous paper [7] in detail. The acetone process [21] is a process mainly used to control the viscosity of PU polymer during the polymerization. Acetone process is often used because it is inert with the WPU forming reactions, can be mixed with water, and has a low boiling point. The advantage of acetone process includes not only obtaining a homogeneous solution but also wide range of structure and emulsion, high quality and reliable reproducibility of end products. 

Subsequently, the 1.5 g amount of RCNs was added with different ways in the WPU synthetic process, which was shown in Scheme 1. In the first method, the RCNs DMF suspension was used, of which suspension was done from water to acetone then from acetone to DMF by several successive centrifugation steps at 12,000 rpm and 10 °C for 20 min. The dispersion was added after the IPDI addition process, stirred for 2 h, and the reaction was allowed to continue for 2 more h. And then DMF removed by vacuum between 60 to 80 °C, which was coded as RCNs/WPU-DMF. In the second method, the RCNs acetone suspension was used, of which suspension was done from water to acetone by several successive centrifugation steps as same as first method of RCNs/WPU-DMF. Finally, a stable suspension in acetone was obtained through 10 min ultrasonic treatment. It was added to the acetone pouring step of the WPU synthesis, which was coded as RCNs/WPU-acetone. In the third method, the RCNs aqueous suspension was added instead of pure water during the dispersion step of the pre-polymer. It coded as RCNs/WPU-water. The WPU prepolymer molecules with isocyanate groups have cross-linked effectively with the hydroxyl groups on the RCNs. 

### 2.4. Preparation of RCNs/WPU Nanocomposite Films

RCNs/WPU nanocomposite films were prepared by casting process. RCNs/WPU aqueous dispersion was sonicated for 1 h before casting and poured on glass plate, which was set square molding type by Teflon tape with 0.32 mm thickness. Then drying was conducted in a chamber at 25 °C and 50% relative humidity for 24 h, and it was progressed in oven at 80 °C for 1 h and in vacuum oven at 40 °C for 24 h, respectively. The thickness of the film used for the tensile strength was 0.16 ± 0.02 mm.

## 3. Analysis of RCNs/WPU Composite Solutions and Films

Transmission electron microscope (TEM): Samples were sonicated for 30 min and completely dispersed in distilled water. The droplets of the sample dispersion were deposited on a polycarbonate film supported on a copper lattice. After drying at room temperature, 1 × 10^−9^ (Gun), 8.82 × 10^−8^ (Column) Torr and air, a transmission electron microscope (JEM-2010, Jeol, Tokyo, Japan) was collected at an acceleration voltage of 200 kV.

Brookfiled viscosity: The viscosity of the WPU and RCNs/WPU dispersions was measured by a DV2T viscometer (Brookfiled, Middleborough, MA, USA). Experiments were carreid out at 12, 30, or 60 rpm of the spindle at 25 °C, respectively.

Zata-potential analysis: The zata potential of the dispersion was measured by diluting the dispersion to 0.5 wt % with deionized water before measurement and then measuring the electrical potential at 25 °C with a Nano-ZS 90 zata potential analyzer (Malvern Instrument Co., Ltd, Worcester, UK) Respectively.

Particle size distribution: The average particle size and the its distribution of the WPU were measured using a Nano-ZS 90 zata-potential analyzer (Malvern Instrument Co., Ltd, Worcester, UK). 0.1 mL of the WPU was diluted with 3 mL deionized water. The sample was added to a deionised water tank the pinhole of 200 μm and measured at 25 °C.

Fourier transform infrared analyses (FT-IR): FTIR spectroscopy was observed at room temperature on a Nicolet FTIR spectrometer (Perkin-Elmer, Waltham, MA, USA). Powdered RCNs and RCN/WPU were crushed with KBr and compressed to produce pellets. 

UV-vis absorption spectrum: Transmittance of WPU, RCNs/WPU nanocomposite was measured by UV-vis spectrometer (SHIMADZU UV-3600, Tokyo, Japan) with the scan wavelength from 800 nm to 400 nm and 1nm per step. The film was shaped to 10 mm × 50 mm × 1 mm. 

Thermogravimetric analysis (TGA): Thermogravimeter (TGA Q500, TA instruments, New Castle, DE, USA) was used to measure the weight loss of the RCNs, WPU, and RCN/WPU nanocomposite under a nitrogen atmosphere. The samples were heated from 30 to 600 °C at a heating rate of 10 °C/min. Generally, 10 mg samples were used for thermogravimetric analysis.

Dynamic mechanical analysis (DMA): The dynamic mechanical thermal properties of the film samples were measured at 1 Hz with a DMA-Q800 (TA Instruments, New Castle, DE, USA) at a heating rate of 4 °C /min from −80 to 80 °C. The size of the films sample were 20 mm × 6 mm × 0.2 mm for the DMA measurements. 

Measurements of mechanical properties: The tensile strength was measured by Autograph tester (Instron 4201, Sidmazu, Tokyo, Japan). The width and length of the samples were set to 5 and 20 mm, respectively, with dumbbell shape, respectively, and the test speed was set to 100 mm/min. 

## 4. Results and Discussion

### 4.1. Properties of RCNs Nanoparticles 

Figure 1 shows TEM images of regenerated cellulose nanoparticles (RCNs). They are not a rod-like cellulose nanocrystal from H_2_SO_4_ but spherical shape which was regenerated from NaOH/urea system. The role of hydroxyl ions of NaOH is described in previous paper [7] 

This difference was leads to either reduction in intensity or increase of certain IR bands characteristic of cellulose crystalline domains or considered as an amorphous band. Figure 2 shows FTIR spectroscopy to explain difference in degree of crystallinity between RCNs and cellulose nanocrystals (CNCs) from H_2_SO_4_ method. The FTIR absorption band at 1430 and 893 cm^−1^ were correspond to CH_2_ bending vibration and C–O–C stretching vibration at β-(1,4)glycosidic linkage, they are known as the crystallinity band and amorphous band in cellulose, respectively [19]. A significant reduction in the intensity of the crystallinity band (1430 cm^−1^)and clearly increased in the intensity of the amorphous band(893 cm^−1^) in the FTIR spectrum of RCNs suggests that the crystallinity of RCNs lower than that of CNCs. The crystallinity of RCNs and CNCs was quantitatively compared by calculating the crystallinity index of RCNs and CNCs using following Equation [22].
CrR = A_1430_/A_893_(1)
where, A1430 and A893 correspond to absorbance at 1430 cm^−1^ (crystallinity band) and 893 cm^−1^ (amorphous band), respectively. The calculated values of crystallinity index for RCNs and CNCs were 0.5 and 0.87, respectively. The significant reduction in the intensity of FTIR bands at 1055, 1107, and 1161 cm^−1^ in RCNs was found, which corresponds to reduction inter- and intra- molecular hydrogen bonding in RCNs. A newly appeared bands at 993 cm^−1^, observed in amorphous cellulose [22], was found not in CNCs but RCNs. 

### 4.2. Characteristics of the RCNs/WPU Nanocomposites Dispersions 

The type of polyurethane water dispersion varies depending on the ionic bond that can be generated when mixing the ionic additive in the post-treatment process. An important and practical type of waterborne PU (WPU) is the anionic type. The WPU has a pendant ionized carboxylic acid group and is synthesized by a four step process as follows. First, the macromonomer diisocyanate is prepared by reacting the excess diisocyanate with a long chain polyol. Then carboxylic acid-containing macromonomer diisocyanate is prepared through dimethylolpropionic acid (DMPA) is incorporated into the backbone of macromonomer diisocyanate. The next step, carboxylic acid was neutralized with tertiary amine. Finally, the anionic WPU prepolymer is vigorously stirred in water. The chain extenders transform the residual isocyanate groups in water into urea bonds to produce an anionic WPU that is stably dispersed. In synthesizing WPU, water plays the role of chain extender to react with terminal isocyanate groups. For this reason, WPU synthesized without addition of a chain extender may be used in eco-friendly or biodegradable applications [5]. 

Based this procedure, the RCNs were added as above shown in Scheme 2 (right) (case of RCNs/WPU-DMF and RCNs/WPU-Acetone). In the liquid suspension, interfacial polar interactions are possible between the RCNs molecules and the liquid polymeric components. There are also strong H-bonding forces between themselves. The low viscosity of the liquid medium allows the RCNs molecules to move due to thermal fluctuations. The molecules can spatially rearrange, and form large structures as shown in Scheme 2 (left). The dispersions states and properties are shown in Table 1. 

The mean particle size distribution of the WPU dispersions was 322 nm, and that of RCNs/WPU nanocomposites was larger than that of WPU. The particle formation was affected by amount of residual free NCO groups, which can react with added water to result in urea group and larger particles [23]. The orders of average particle size for nanocomposite dispersions were RCNs/WPU-DMF, RCNs/WPU-Acetone and RCNs/WPU-Water. This change could results from the ionic and hydrogen bonding interactions between WPU and nanocellulose particles contains much amorphous regions. 

The WPU aqueous dispersion has a negative zeta potential of −49.5 mV. The particles of the WPU dispersion were stabilized in water by COO– ions of neutralized DMPA. Thus, the WPU particles represent negative surface charge and zeta potential. The zeta potentials of the nanocomposites dispersions become less negative due to the addition of nanocellulose particles. The hydroxyl group of nanocellulose would interact with NCO– of WPU pre-polymer chain end and urethane bond was formed. The dispersion of RCNs/WPU-DMF shows the lowest particle size distribution, followed by RCNs/WPU-Acetone, and RCNs/WPU-Water.

The Brookfield viscosity of the RCNs/WPU nanocomposites was higher than that of WPU due to its nanocomposites between WPU pre-polymers and nanocellulose particles. With higher mean particle size, the viscosity of nanocomposites was increased in the order DMF, acetone, and water-dispersed. In case of RCNs/WPU-Water, the nanocelluose particles were inserted in last step of synthesis process of WPU, which acts as chain extender between WPU pre-polymer in synthesis. This produced increasing the mean particle size of RCNs/WPU nanocomposites. 

Figure 3 shows IR spectroscopy of WPU, RCNs/WPU-DMF, RCNs/WPU-Acetone, and RCNs/WPU-Water. In the WPU+RCNs spectrums, we could observed the stronger appearance of composited WPU spectrum than neat WPU at around 1731, 1664, 1365, 1307, 1196, and 1062 cm^−1^, which were associated with the absorption of urethane hydrogen-bonded carbonyl groups (C=O), urea hydrogen-bonded carbonyl groups (C=O), cellulose CH stretching band, cellulose CH_2_ stretching band, cellulose C–C ring vibration, and cellulose C–O–C ring vibration [24], respectively. The results indicated in Table 2. 

The nanocomposite WPU + RCNS can prove very efficient dispersion of cellulose nanoparticles by IR results. Pei et al. [25] has already demonstrated covalent bond formation between cellulose nanocrystals and polyurethane through FTIR spectra. The absorption of the carbonyl group (C=O) is judged to be due to the reaction of hydroxyl group of cellulose and isocyanate group of WPU. The peak intensity and area about hydrogen bond of WPU was order of RCNs/WPU-DMF > RCNs/WPU-Acetone > RCNs/WPU-Water. The RCNs/WPU-DMF was most effective method for in-situ polymerization of WPU. From the nanocomposite with nanocellulose, however the RCNs/WPU-Acetone appeared to be more effective, which was estimated the intensity of related nanocellulose bands at 1302, 1196, and 1664 cm^−1^. From the RCNs/WPU manufacturing process point of view, the method RCNs/WPU-Acetone is simpler than the method RCNs/WPU-DMF, of which polymerization method is troublesome to evaporate DMF at an intermediate stage. 

### 4.3. Characteristics of the RCNs/WPU Nanocomposites Films 

Transparency: The UV-Vis transmission spectra of the different composite films were tested and the results are shown in Figure 4. The transmittance of WPU film was highest in the visible region with a value of about 88.8% at 600 nm. The RCNs/WPU composite films have poorer visible light transmittance than the WPU film. The order of transmittance were WPU > RCNs/WPU-DMF > RCNs/WPU-Acetone > RCNs/WPU-Water. This result is related to tendency of mean particle size of RCNs/WPU nanocomposites. With increase particle size, the film looks opaque. 

Tensile, dynamic mechanical properties: Static / cyclic loading and dynamic mechanical analysis (DMA) tensile tests were conducted to study the toughness and softening of RCNs/WPU nanocomposites. The stress–strain (S−S) curves of the WPU and RCNs/WPU nanocomposites are shown in Figure 5A. 

The resulting values, such as the tensile strength, strain-to-failure, and Young’s modulus are summarized in Table 3. There is a remarkable development in the tensile strength associated with increasing strain simultaneously. In the case of the neat WPU, it had an S−S curve with 800% strain and 7.8 MPa tensile strength at break, the RCNs/WPU nanocomposites containing 1 wt % RCNs exhibits stronger stress and higher elongations than WPU film. The tensile strength increases from 7.8 MPa for neat WPU to 27.9 MPa for RCNs/WPU-DMF or RCNs/WPU-Acetone. This is more than 4 times the neat WPU value. The strain value for the RCNs/WPU nanocomposites reaches the highest value of 1235% for RCNs/WPU-DMF with containing 1 wt % RCNs, which value is almost 2 times higher than that of neat WPU. There is a tendency that the strain of the nanocomposite is further increased after manufacture of RCNs nanocomposites. The Young’s modulus of the RCNs/WPU nanocomposites increases to 10.7 MPa for RCNs/WPU-Water compare to 1.8 MPa for the neat WPU. This increase is due to the stiffening of the polymers. Polyurethane is block copolymer with alternating soft and hard segments. In this study, the blocks of the WPUs consist of IPDI and PCDL, which are role as hard and soft segments, respectively. The soft phases are derived from the polyols linked by IPDI and roles of elastomeric properties. The hard phase for the IPDI segment with high polar urethane bonds is bounded by the nanocellulose particles. The domains in which these nanoparticles are introduced acts as physical crosslinking agent and acts as high modulus filler. The nanocellulose particles were introduced in the reactor before increasing viscosity (DMF-dispersed type) of the pre-polymer; at this step, there are free NCO groups at the ends of the low molecular polyurethane chains. This result explained that later addition of nanocellulose of DMF-dispersed steps induces an increase in the effective cross-link by homogeneous dispersion of the nanoparticles in the nanocomposites than Acetone and Water-dispersed step. This linkage can be attributed to new interactions formed between WPU and RCNs. For the Acetone dispersion step, the nanocellulose was added to the viscosity control process of the polymer with acetone, which is finished formation of pre-polymer of polyurethane. Nanocellulose particles have many hydroxyl groups on surface, which roles as graft-point of polyurethane chains as Scheme 2. In case of Water-dispersed step, the nanocelluloses were introduced during aqueous dispersion step of the pre-polymer, which roles as chain extender in WPU synthesis process. This part appears as a hard segment and contributes to the high modulus of the WPU composite matrix through hydrogen bonding. The role as chain extender of nanocellulose particles for RCNs/WPU synthesis process is mainly formed by Water-dispersed step. Although the effects of reinforcement WPU by cellulose nanoparticles were accomplished, the roles of nanoparticles were likely different in detail according to when they introduced. Comparing with cellulose nanocrystal or nanofibers for reinforcement, semi-crystalline nanocellulose particles improved the strain property of RCNs/WPU without stress property damage.

To understand the elastomeric recovery and deformation of neat WPU and the RCNs/WPU nanocomposites, a cyclic tensile test was conducted. The tensile loading and unloading cycles performed at a maximum strain of 50% with a constant speed at 130 mm·m^−1^. The loops of hysteresis for cycles 20 are shown in Figure 5B. The large hysteresis loop showed unrecoverable deformation in the first cycle, and followed by very small hysteresis with negligible residual strain values in the next 19 cycles. This is desirable phenomenon for an elastomeric material like PU (Table 3). The large hysteresis region of the first loops in the RCNs/WPU nanocomposites represents the stretch-induced softening due to the increased volume fraction of the effective soft phases as a result of conversion from hard domains to soft phases during the initial alignment of the microstructures [26]. However, RCNs/WPU-Water and RCNs/WPU-Acetone displays a decrease in softening comparing to RCNs/WPU-DMF, as shown in Figure 5B. This result can be explained that entanglement and the networks from addition of RCNs in hard segments, which induced enhancing mechanical properties of the RCNs/WPU nanocomposite. These results are in good agreement with the tendency of the Young’s modulus (E), as shown in Table 3.

Dynamic mechanical analysis (DMA) of the WPU and RCNs/WPU nanocomposite were demonstrated that the effect of the temperature and the frequency on the dynamic elastic properties. The storage modulus of the samples is shown in Figure 6. The modulus of the RCNs/WPU nanocomposites are slightly higher than that of the neat WPU at room temperature (20 °C), which increases from 0.17 MPa for neat WPU to 0.37 MPa for RCNs/WPU-DMF. The hardening mechanism is attributed to increasing the hard segments in the RCNs/WPU nanocomposites and effective cross-link density of the RCNs in the nanocomposites. The neat WPU showed a sharply decreased modulus during the glass-rubber transition. However, the decreasing tendency of storage modulus for the RCNs/WPU nanocomposite is slowly reduced because of the confined network structure, providing remarkable thermomechanical stability up to 50 °C. Also, the T_g_ values of the WPU soft segments represents the degree of relaxation of the PCDL phases, which shifted to a lower temperature from −37.3 (neat WPU) to −44.6 °C (RCNs/WPU-DMF). This phenomenon is incompatible with the result of complexing with highly crystalline nanocellulose [25] Generally, when highly crystalline nanoparticles (CNs) are introduced, the hydrogen bonded carbonyl groups among the hard segments formation increased and strong association with CNs of the hard segments [25]. However, since the RCN has a semi-crystalline phase, it maintains a soft chain even in the hard segment of WPU, so that the damping due to the external environment is formed at a lower temperature than neat WPU. This means that the cross-coupling between the WPU molecule and the RCN increases the movement of the WPU chain, thereby increasing the damping capacity.

Thermal properties: Figure 7 shows the TGA curves for WPU, RCNs and RCNs/WPU nanocomposite with different loadings steps of cellulose nanoparticles. The thermogram of RCNs shows 2 steps at around 150–300 °C and 350–400 °C, which were corresponding to dehydration and decomposition of glycosyl units, and oxidation and breakdown to lower molecular weight gaseous products respectively, followed by the formation of a char. The curve of WPU shows 2 steps, first decomposition stage (from approximately 280–320 °C) results from urethane/urea-bond degradation of hard-segments; second decomposition stage (from approximately 320–380 °C) corresponding of polyol degradation of soft-segments. The RCNs/WPU nanocomposite have 2 steps degradation at 280–400 °C due to decomposition of nanocellulose particles as chain extender in hard segment in RCNs/WPU nanocomposite and a maximum degradation at about 400 °C over due to WPU chains, which is much higher than the degradation temperature of pure WPU (320–380 °C). This significant enhancement of thermal resistance by the presence of RCNs can be attributed to the formation of a confined structure in the RCNs/WPU nanocomposites. Not only the T_max_ of RCNs/WPU nanocomposite is higher than that of WPU, but also the start temperature of decomposition is different with loadings steps of cellulose nanoparticles. The difference between RCNs-Acetone and RCNs-Water was not significant, but RCNs-DMF showed higher thermal stability than other samples. The role of nanocellulose particles as thermos-stability enforcement of nanocomposites is most effect in RCNs/WPU-DMF.

## 5. Conclusions

In this study, RCNs with improving solubility and stability of solution, thereby resulting in lower crystallinity were fabricated by using a NaOH-based solution with urea. Additionally, the stronger chemical bond between RCNs and WPU was here founded by investigating which stage in particular added by RCNs worked best on strengthening their bond. Finally, we could develop RCNs reinforced WPU nanocomposites with biodegradability as well as enhancing mechanical properties through a facile, eco-friendly approach fittable for global politics. FTIR indicated that the RCNs/WPU nanocomposites using a DMF dispersed step has the strongest intensity of hydrogen bonds between cellulose and isocyanate groups attributed by the absorption of carbonyl groups (C=O). Tensile test also proved that the DMF-dispersed step induce an increase in the effective crosslink density by homogeneous dispersion of the nanofillers in the nanocomposites, with 27.9 MPa of 4 times more improvement value, 1235% of strain-to-failure, 2 times higher value. The moduli were also higher than that of the neat WPU matrix illustrating that the value of G’ of the nanocomposites increases from 0.17 MPa for neat WPU to 0.37 MPa for RCNs/WPU-DMF, and the T_g_ of the WPU soft segments were shifted to a lower temperature from −37.3 to −44.6 °C for the neat WPU and RCNs/WPU-DMF, because the breaking of hydrogen bonded carbonyl groups in the hard segments being essential for inhibiting of HD formation, accordingly causing the enhancement in the mobility of the poly(ether) chain in RCNs/WPU, comparing with the neat WPU sample. TGA results displayed that the role of RCNs as thermos-stability enforcement of the resultant nanocomposites was the most effective in the DMF-dispersed step. 

In conclusion, the above all results signified that the properties involving the mechanical, thermal, and chemical specific features, of RCNs/WPU nanocomposites by using the DMF-dispersed step were remarkably improved, comparing with acetone or water-dispersed steps. Therefore, fabricated nanocomposites have a great potential for further applications such as environmentally friendly artificial leather processes, coatings, and biomedical fields.

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
