# Peer review of "One-Pot Processing of Regenerated Cellulose Nanoparticles/Waterborne Polyurethane Nanocomposite for Eco-friendly Polyurethane Matrix"

_polymers, 2019, doi:10.3390/polym11020356_

Round 1

Reviewer 1 Report

This manuscript is based on preparation and characterization of regenerated cellulose nanoparticles (RCNs) reinforced waterborne polyurethanes (WPU). This reinforcement was done to enhance mechanical properties and other final properties of design nanocomposites. The chemical structure, mechanical, particle size and distribution, viscosity, and thermal properties of the resultant RCNs/WPU nanocomposites were investigated by Fourier transform infrared analysis (FTIR), Zeta-potential analysis, viscometer, thermogravimetric analysis (TGA), universal machine Instron and dynamic mechanical analysis (DMA). This paper is interesting and should be publish after minor revision.

Minor revision:

Line 126: “normal temperature, normal pressure” should be change to standard temperature and pressure or ambient room temperature and standard pressure

Line 170: the text “crystallinity band” and “amorphous band” should be written without “”

Line 436: please revised reference 23, this is the author name and not author surname.

Author Response

Response to Reviewer #1

This manuscript is based on preparation and characterization of regenerated cellulose nanoparticles (RCNs) reinforced waterborne polyurethanes (WPU). This reinforcement was done to enhance mechanical properties and other final properties of design nanocomposites. The chemical structure, mechanical, particle size and distribution, viscosity, and thermal properties of the resultant RCNs/WPU nanocomposites were investigated by Fourier transform infrared analysis (FTIR), Zeta-potential analysis, viscometer, thermogravimetric analysis (TGA), universal machine Instron and dynamic mechanical analysis (DMA). This paper is interesting and should be publish after minor revision.

We greatly appreciate the reviewer’s comments.

Comment 1:

Line 126: “normal   temperature, normal pressure” should be change to standard temperature and   pressure or ambient room temperature and standard pressure

→ According to the reviewer’s indication, we corrected and added explanation in the part of preparation of 5 page.

“After drying at normal temperature, normal pressure and air, a transmission electron microscope (JEM-2010, Jeol, Jepan) was collected at an acceleration voltage of 200 kV.”

was changed to:

“After drying at room temperature, normal pressure (X torr) and air, a transmission electron microscope (JEM-2010, Jeol, Jepan) was collected at an acceleration voltage of 200 kV.”

Comment 2:

Line 170: the text   “crystallinity band” and “amorphous band” should be written without “”

→ According to the reviewer’s indication, we corrected it.

“The FTIR absorption band at 1430 cm-1 and 893 cm-1 were correspond to CH2 bending vibration and C-O-C stretching vibration at β-(1,4) glycosidic linkage, they are known as the “crystallinity band” and “amorphous band” in cellulose, respectively [19].”

was changed to:

“The FTIR absorption band at 1430 cm-1 and 893 cm-1 were correspond to CH2 bending vibration and C-O-C stretching vibration at β-(1,4) glycosidic linkage, they are known as the crystallinity band and amorphous band in cellulose, respectively [19].”

Comment 3:

Line 436: please revised   reference 23, this is the author name and not author surname

→ According to the reviewer’s indication, we corrected it in 13 page.

“Arantzazu, S.E.; Lorena, U.; Aitor A.; Filomena, B.; Maria A.C.; Arantxa, E. Modulating the microstructure of waterborne polyurethanes for preparation of environmentally friendly nanocomposites by incorporating cellulose nanocrystals, Cellulose 2017, 24, 823–834.

was changed to:

“Santamaria-Echart, A.; Ugarte, L.; Arbelaiz, A.; Barreiro, F.; Corcuera, M.A.; Eceiza, A. Modulating the microstructure of waterborne polyurethanes for preparation of environmentally friendly nanocomposites by incorporating cellulose nanocrystals, Cellulose 2017, 24, 823–834.

Reviewer 2 Report

Comments to the Author

This work reported regenerated cellulose nanoparticles (RCNs) reinforced waterborne polyurethanes (WPU) were developed to improve mechanical properties as well as biodegradability by using a facile, eco-friendly approach. This work obtained some interesting results. However, the paper have been carefully discussed. Major revision should be done before publication. The following comments are listed for improvement of the paper:

1.    It is mentioned in the 97th row that “it reduces the NCO-NH reactivity that can be formed 98 through reversible ketamine formation”, this sentence is quite ambiguous, it is hoped that this sentience can be explained much more clearly.

2.    In Scheme 1, the position of RCN addition should be expressed, and is it the solid obtained after removing the DMF solvent? If it is, it is necessary to clearly interpret that after the polyurethane is solidified, the acetone solvent is used to dissolve the solid for preparing the second sample, why different experiments are made in different solvent systems?

3.    In Scheme 1, what is the purpose for dispersing the solution at 300-400rpm after adding water? At present, most water-borne PU solution is dispersed at 1000rpm or above to get smaller PU emulsion bubbles, in order to obtain better dispersion result.

4.    Is the molecular weight measured in this paper? Because the water in DMF or acetone may cause that the molecular weight of polyurethane is unable to be raised, and the molecular weight will influence the properties of water-borne polyurethane.

5.    In Scheme 2, is DMPA able to react with PCDL actually in the research condition of this paper, is it necessary to be changed to IPDI-DMPA-IPDI-PCDL-IPDI.

6.    The figures in Scheme 3 and Scheme 2 should be combined into a figure, in order to make the paper more succinctly.

7. In the paper, it is described that there is stronger chemical bond between RCNs and polyurethane, is there any clearer evidence reveals that there is stronger chemical bond between RCNs and polyurethane.

    8. Is XRD is tested to prove its crystallinity more effectively?

9. What is the dispersion property of WPU in RCNs? Is the TEM of RCNs adding in WPU tested?

   10. In the conclusion, it is mentioned in the 363th row that “RCNs with     improving solubility and stability of solution”, but there is no tests               conducted for solubility and stability of solution in the paper.

11. In the conclusion, it is mentioned in the 367th row that “reinforced WPU nanocomposites with biodegradability as well as enhancing…”, but there is no test conducted for biodegradability in the paper.

Author Response

Response to Reviewer #2

This work reported regenerated cellulose nanoparticles (RCNs) reinforced waterborne polyurethanes (WPU) were developed to improve mechanical properties as well as biodegradability by using a facile, eco-friendly approach. This work obtained some interesting results. However, the paper have been carefully discussed. Major revision should be done before publication. The following comments are listed for improvement of the paper:

We greatly appreciate the reviewer’s comments.

Comment 1:

It is mentioned in the   97th row that “it reduces the NCO-NH reactivity that can be formed 98 through   reversible ketamine formation”, this sentence is quite ambiguous, it is hoped   that this sentience can be explained much more clearly.

→ According to the reviewer’s indication, we considered to delete the sentence, because of the mechanism of chain extension in case of amine groups in solvent.

“Acetone process is often used because it is inert with the WPU forming reactions, can be mixed with water, and has a low boiling point. In addition, it reduces the NCO-NH reactivity that can be formed through reversible ketamine formation. The advantage of acetone process includes not only obtaining a homogeneous solution but also wide range of structure and emulsion,~”

was changed to:

“Acetone process is often used because it is inert with the WPU forming reactions, can be mixed with water, and has a low boiling point. In addition, it reduces the NCO-NH reactivity that can be formed through reversible ketamine formation. The advantage of acetone process includes not only obtaining a homogeneous solution but also wide range of structure and emulsion,~”

Comment 2:

In Scheme 1, the   position of RCN addition should be expressed, and is it the solid obtained   after removing the DMF solvent? If it is, it is necessary to clearly   interpret that after the polyurethane is solidified, the acetone solvent is   used to dissolve the solid for preparing the second sample, why different   experiments are made in different solvent systems?

We prepared 1 control (no RCNs) and 3 samples (including RCNs) with different RCNs adding sequence. According to the adding sequence, the samples were named following as.

WPU                  : no RCNs

RCNs/WPU-DMF       : RCNs dispersed in DMF

RCNs/WPU-Acetone     : RCNs dispersed in Acetone

RCNs/WPU-Water       : RCNs dispersed in Water

The sample (RCNs/WPU-DMF) was prepared with adding RCNs dispersed in DMF. DMF was used only for adding RCNs to WPU polymer precursor solution, so then, DMF was removed by vacuum drying at 60 °C. Therefore, the RCNs were still dispersed in WPU polymer precursor solution. 

This article is about WPU(waterborne polyurethane), not PU(polyurethane). There are many references (J. Appl. Polym. Sci. 2018, 135, 46633-46641., Aqueous polyurethane dispersions. Colloid. Polym. Sci. 1996, 274, 599-611.)

Comment 3:

In Scheme 1, what is the   purpose for dispersing the solution at 300-400rpm after adding water? At   present, most water-borne PU solution is dispersed at 1000rpm or above to get   smaller PU emulsion bubbles, in order to obtain better dispersion result.

→ Immediately after adding water, the suspension was stirred at 1000rpm, but no need to continuously keep high speed. So, one hour later, it retains at 300-400rpm. We did not have any problems with dispersion.

Comment 4:

Is the molecular weight   measured in this paper? Because the water in DMF or acetone may cause that   the molecular weight of polyurethane is unable to be raised, and the   molecular weight will influence the properties of water-borne polyurethane.

→ when we synthesized only WPU, we measured the molecular weight. However, we did not measure the molecular weight of WPU with RCNs. The main purpose was not to increase the molecular weight, due to the application of developed WPU/RCNs for medical area.   

Comment 5:

In Scheme 2, is DMPA able to react   with PCDL actually in the research condition of this paper, is it necessary   to be changed to IPDI-DMPA-IPDI-PCDL-IPDI.

→ The procedure of that DMPA is incorporated into the backbone of macromonomer diisocyanate was well presented on a reference [6] (Adv. Exp. Med. Biol. 2018, 1077, 251-283.).

“The most important and practical type of waterborne PU is the anionic type. This type of waterborne PU possesses pendant ionized carboxylic acid groups. Anionic waterborne PUs with carboxylic acid groups can be synthesized by a four-step process, In the first step, macromonomer diisocyante is prepared by reacting excess diisocyante with a long-chain polyol and/or low molecular weight glycol. Then carboxylic acid-containing macromonomer diisocyanate is prepared through the hydrophilization of macromonomer diisocyante in the second step, where bis-hydroxycarboxylic acid, such as dimethylolpropionic acid (DMPA), is incorporated into the backbone of macromonomer diisocyante. The next step involves the neutralization of carboxylic acid with tertiary amine. Finally, the anionic PU prepolymer is vigorously sheared and stirred in water diamine. Chain extension in water causes the residual isocyanate group to transform into urea linkage resulting in an anionic PU that is stably dispersed in water.”

Comment 6:

The figures in Scheme 3 and Scheme 2   should be combined into a figure, in order to make the paper more succinctly.

→ According to the reviewer’s indication, we corrected it in 6 page.

Scheme 2. Final dispersion of prepolymer of WPU/RCNs in water (left) and Dispersion of prepolymer of WPU/RCNs in water/acetone (right).

“Based this procedure, the RCNs were added as above shown in Scheme 2 (case of RCNs/WPU-DMF and RCNs/WPU-Acetone). In the liquid suspension, interfacial polar interactions are possible between the RCNs molecules and the liquid polymeric components. There are also strong H-bonding forces between themselves. The low viscosity of the liquid medium allows the RCNs molecules to move due to thermal fluctuations. The molecules can spatially rearrange, and form large structures as shown in Scheme 3. The dispersions states and properties are shown in Table 1.”

was changed to:

“Based this procedure, the RCNs were added as above shown in Scheme 2 (right) (case of RCNs/WPU-DMF and RCNs/WPU-Acetone). In the liquid suspension, interfacial polar interactions are possible between the RCNs molecules and the liquid polymeric components. There are also strong H-bonding forces between themselves. The low viscosity of the liquid medium allows the RCNs molecules to move due to thermal fluctuations. The molecules can spatially rearrange, and form large structures as shown in Scheme 2 (left). The dispersions states and properties are shown in Table 1.”

Comment 7:

In the paper, it is described that there   is stronger chemical bond between RCNs and polyurethane, is there any clearer   evidence reveals that there is stronger chemical bond between RCNs and   polyurethane.

→ We concluded the formation of chemical bond between RCNs and WPU through FTIR spectra with increased intensity of the RCNs functional groups’ peaks in RCNs/WPU.

Comment 8:

Is XRD is tested to prove its   crystallinity more effectively?

→ Though XRD was tested, the separation of peaks was embarrassing owing to observations with relatively large amorphous peak of WPU. Because, RCNs are semi crystal polymer, which added with tiny portion into WPU and WPUs are also amorphous polymer. In reference [7]( J. Appl. Polym. Sci. 2018, 135, 46633-46641.), increasing the amount of RCNs, the difference of peak was shown, however the properties were declined. Therefore, we have included RCN in the WPU to the extent that there is no deterioration of the properties of the WPU.

Comment 9:

What is the dispersion property of   WPU in RCNs? Is the TEM of RCNs adding in WPU tested?

→ RCNs were as a disperse phase within WPU. The SEM image of RCNs/WPU was displayed in reference [7]( J. Appl. Polym. Sci. 2018, 135, 46633-46641.)

Comment 10:

In the conclusion, it is mentioned   in the 363th row that “RCNs with improving solubility and stability of   solution”, but there is no tests conducted for solubility and stability of   solution in the paper.

→ We indirectly concluded through the enhancement of mechanical and thermal properties. If the solubility and stability were insufficiency, the properties could not increase.

Comment 1 1:

In the conclusion, it is mentioned   in the 367th row that “reinforced WPU nanocomposites with biodegradability as   well as enhancing…”, but there is no test conducted for biodegradability in   the paper.

→ It was mentioned in reference [7]( J. Appl. Polym. Sci. 2018, 135, 46633-46641.)

Round 2

Reviewer 2 Report

Dear editor:

The manuscript can be accepted and published in polymers.